# Specific Mutations in Aph1 Cause γ-Secretase Activation

**DOI:** 10.3390/ijms23010507

**Published:** 2022-01-03

**Authors:** Hikari Watanabe, Chika Yoshida, Masafumi Hidaka, Tomohisa Ogawa, Taisuke Tomita, Eugene Futai

**Affiliations:** 1Laboratory of Enzymology, Graduate School of Agricultural Sciences, Tohoku University, Sendai 980-0845, Japan; hikari.watanabe.q6@dc.tohoku.ac.jp (H.W.); chikayoshida2015@gmail.com (C.Y.); masafumi.hidaka.a4@tohoku.ac.jp (M.H.); tomohisa.ogawa.c3@tohoku.ac.jp (T.O.); 2Laboratory of Neuropathology and Neuroscience, Graduate School of Pharmaceutical Sciences, The University of Tokyo, Bunkyo-ku, Tokyo 113-0033, Japan; taisuke@mol.f.u-tokyo.ac.jp

**Keywords:** Alzheimer’s disease, amyloid beta (Aβ), gamma-secretase, Aph1, intramembrane proteolysis, *Saccharomyces cerevisiae*

## Abstract

Amyloid beta peptides (Aβs) are generated from amyloid precursor protein (APP) through multiple cleavage steps mediated by γ-secretase, including endoproteolysis and carboxypeptidase-like trimming. The generation of neurotoxic Aβ42/43 species is enhanced by familial Alzheimer’s disease (FAD) mutations within the catalytic subunit of γ-secretase, presenilin 1 (PS1). FAD mutations of PS1 cause partial loss-of-function and decrease the cleavage activity. Activating mutations, which have the opposite effect of FAD mutations, are important for studying Aβ production. Aph1 is a regulatory subunit of γ-secretase; it is presumed to function as a scaffold of the complex. In this study, we identified Aph1 mutations that are active in the absence of nicastrin (NCT) using a yeast γ-secretase assay. We analyzed these Aph1 mutations in the presence of NCT; we found that the L30F/T164A mutation is activating. When introduced in mouse embryonic fibroblasts, the mutation enhanced cleavage. The Aph1 mutants produced more short and long Aβs than did the wild-type Aph1, without an apparent modulatory function. The mutants did not change the amount of γ-secretase complex, suggesting that L30F/T164A enhances catalytic activity. Our results provide insights into the regulatory function of Aph1 in γ-secretase activity.

## 1. Introduction

γ-Secretase is an intramembrane-cleaving protease responsible for generating the amyloid beta peptide (Aβ), the accumulation and deposition of which is a major pathological hallmark of Alzheimer’s disease (AD) [1]. Aβ is generated from amyloid precursor protein (APP) by sequential cleavages of β- and γ-secretases [2,3]. In the first step, the extracellular portion of APP is cleaved by β-secretase, known as BACE1, to generate a β-carboxyl terminal fragment (β-CTF). γ-Secretase cleaves β-CTF in its transmembrane domain through two different protease activities [3]. An initial endopeptidase-like cleavage of β-CTF releases the APP intracellular domain (AICD), giving rise to Aβ48 or Aβ49. Further stepwise carboxypeptidase-like trimming occurs to produce major Aβ species composed of 38–43 amino acids [4,5]. The longer Aβ42 and Aβ43 species are highly aggregation-prone and neurotoxic, compared with the shorter Aβs (Aβ38 and Aβ40) [1,6]. An increase in the ratio of Aβ42/43 species leads to the accumulation of Aβ fibrils and the deposition of senile plaques in the brain, resulting in AD [7,8]. Recently, γ-secretase modulators (GSM) were found to alter the ratios of Aβ species, particularly decreasing the ratios of longer Aβs [3]. Small molecules reducing toxic Aβ42/43 have been proposed as a potential cure, and GSM is a potential therapeutic agent for the treatment of AD.

γ-Secretase is a protein complex composed of four membrane proteins: presenilin (PS1 or PS2), nicastrin (NCT), anterior pharynx-1 (Aph1), and presenilin enhancer 2 (Pen2) [9]. PS1 possesses nine transmembrane domains (TMD1–9); TMD6 and TMD7 contain two catalytic aspartic acids [10,11]. Upon assembly and maturation of the complex, PS1 is cleaved in the loop between TMD6 and TMD7, thereby generating N-terminal fragments (NTFs) composed of TMD1–6 and C-terminal fragments (CTFs) composed of TMD7–9 [3]. More than 150 pathological mutations in PS1 are associated with familial Alzheimer’s disease (FAD) [1]. These clinical PS1 mutations negatively affect γ-secretase activity, resulting in a decrease in Aβ generation. Decreased trimming activity leads to increases in the ratios of Aβ42 or Aβ43, which are responsible for AD [12,13]. The four transmembrane domains (TMD1–4) of Aph1 interact with PS1 and NCT [14]. To form the γ-secretase complex, Aph1 first binds NCT; it then binds PS1 and Pen2 [9]. Thus, Aph1 has an important role as a scaffold protein and stabilizes PS1 holoprotein in the complex, whereas Pen2 is required for endoproteolytic processing of PS1 and conferring γ-secretase activity of the complex [9]. Two Aph1 homologs, Aph1a and Aph1b, have been identified in humans. The γ-secretase complex containing Aph1b has been shown to produce more Aβ42 than the complex containing Aph1a, the major homolog; different conformations and functions have been proposed for the two Aph1 homologs [15].

Structural analyses of γ-secretase have provided insight into its cleavage mechanism. Cryo-electron microscopy (EM) of γ-secretase revealed that the transmembrane domains of each subunit are arranged in a horseshoe shape and the extracellular domain of NCT rests on the transmembrane helices [16]. PS1 exhibits a hydrophilic catalytic pore structure, which is water-accessible (according to the findings in biochemical studies) [17,18,19]. Aph1 transmembrane domains are in contact with PS1 transmembrane domains and the C-terminal loop, as well as NCT extracellular and transmembrane domains [20]. Pen2 are in contact with PS1 transmembrane and NCT extracellular domains on the opposite side of the horseshoe. Lateral movement of β-CTF into the pore structure has been proposed as a cleavage mechanism [21]. Upon recruitment of substrates, the extracellular domain of NCT [22,23] and the recognition sites in PS1 interact with the substrates [24]. A cross-linkage study between β-CTF and PS1 revealed that β-CTF first interacts with PS1 NTF; it is then transferred to PS1 CTF for proteolysis [25]. Recent updates of γ-secretase structures in complex with the substrate APP or Notch have revealed the structural basis of γ-secretase recognition [26,27]. PS1 forms a hybrid β-sheet with substrates in the catalytic pore. The hybrid β-sheet pulls the substrate helix and unwinds the cleavage sites of the substrates. In addition, γ-secretase structures in complexes with chemicals have revealed the binding sites of γ-secretase inhibitors (GSI) and GSM [28]: GSI resides in the catalytic pore of PS1 and GSM binds the TMD1 of PS1.

Previously, we established a γ-secretase assay system in yeast, which does not possess homologs of γ-secretase or APP [29]. We detected Aβ production using the Gal4 reporter system and analyzed Aβ species using an in vitro assay with yeast microsomes [29,30]. We identified PS1 mutations that restored the function of PS1 FAD mutants [31]. Suppressor mutations modulated γ-secretase activity and decreased the ratio of Aβ42/43 [31]. Yeast cells are useful to screen for mutations that regulate γ-secretase activity. In the present study, we identified Aph1 mutants that possess protease activity in the absence of NCT. We found that these mutants enhanced γ-secretase endoproteolysis activity with full subunits. We propose that the underlying mutations cause conformational changes in PS1 to enhance protease activity.

## 2. Results

### 2.1. Identification of Aph1 Mutations That Activate γ-Secretase in the Absence of NCT

We used a yeast reconstitution system expressing human γ-secretase subunits and an APP-based substrate, APP_C55_-Gal4p, to screen Aph1aL mutations. Recombinant plasmids were introduced into the yeast strain PJ69-4A, which possesses *HIS3*, *ADE2*, and *lacZ* markers under Gal4p control. When γ-secretase cleaves the substrate, Gal4p is released from membrane-bound APP_C55_-Gal4p and activates the expression of *HIS3*, *ADE2*, and *lacZ*. Therefore, the activity of γ-secretase can be assessed by cell growth in media lacking histidine and adenine, or by β-galactosidase activity [29,31]. In the media, cells could grow with four subunit genes introduced, but not when any single subunit was absent.

We previously identified PS1 mutants active in the absence of NCT [29]. These mutants possessed two missense mutations in PS1: S438P at the ninth transmembrane domain together with one mutation distributed throughout the molecule. These PS1 mutations enhanced and modulated the γ-secretase activity and decreased the ratio of long Aβs [32]. We then isolated Aph1 or Pen2 mutants able to grow in the absence of NCT. After we had screened cells with Pen2 or Aph1 mutations and the PS1 S438P mutation, we found 13 Aph1 mutants that grew in the absence of NCT (Table 1 and Figure 1). Six mutants possessed two mutations and seven mutants possessed one mutation. These Aph1 mutants did not grow with the wild-type PS1 (Table 1), indicating that the PS1 S438P mutation is necessary for the activation of growth without NCT. No Pen2 mutants were active without NCT. These results indicate that PS1 and Aph1 mutations act synergistically to activate γ-secretase in the absence of NCT.

β-Galactosidase activity determined the cleavage of APP_C55_-Gal4p by Aph1 mutants. Aph1 L30F/T164A, L47I/T164A, L47F/V51G, S103P/P216S, V131A/T164A, and I135V/P216S possessed high activity (Figure 2a–f), compared to cells with each replacement. L30F, L47I, S103P, V131A, I135V, T164A, and P216S showed low but significant activities (Figure 2a–f). L47F and V51G showed diminished activities (Figure 2c), suggesting that two mutations are required for full activation. Including mutations regarded as single mutants (Figure 2g), we identified 10 primary mutations that activate γ-secretase (10% to 40% of wild-type Aph1 with NCT), and two secondary mutations that act only in combination (Figure 1).

### 2.2. Aph1 Mutations Activate γ-Secretase Activities

To assess the activities of these Aph1 mutations under physiological conditions in the presence of all γ-secretase subunits, we investigated the Aph1 mutants with wild-type PS1, NCT, and Pen2. The Aph1 mutants L30F/T164A and V131A/T164A showed 1.5-fold greater activity than did the WT (Figure 3a). Immunoblotting confirmed that the expression levels of mutant Aph1 and PS1 were similar to levels in the WT. These results suggest that γ-secretase with these two Aph1 mutations exhibit enhanced protease activity for APP.

### 2.3. Aβ Production by Mutant Aph1 in Yeast Microsomes

We analyzed Aβ-producing activity using the yeast microsomal fraction. The cleavage of APP_C55_ was tested in vitro by incubating the solubilized microsomes (Figure 4) [30]. Aβ was observed in the presence of γ-secretase with the wild-type Aph1 (Figure 4a, lanes 1–4). The amount of Aβ increased in the presence of PC, and diminished with L685,458, a γ-secretase-specific inhibitor. With the Aph1 L30F/T164A mutant, the intensities of the Aβ bands increased in different conditions (Figure 4a, lanes 6 and 7, and Figure 4b), indicating that the mutation enhances Aβ-producing activity. The intensities of different Aβ species, Aβ38, Aβ40, Aβ42, and Aβ43, also increased with L30F/T164A (Figure 4c, lanes 2, 3, 6, and 7, and Figure 4d); the ratio of each species was similar (Figure 4e), except that the ratio of Aβ38 was increased with the mutant. The ratio of long Aβ species (Aβ42 and Aβ43) with Aβ40 was similar. These results suggest that L30F/T164A enhances Aβ production but its Aβ trimming activity is similar to the WT.

One possible explanation for the increased Aβ production in the Aph1 L30F/T164A mutant is that the mutations increase complex formation. To test whether the Aph1 L30F/T164A mutants form a complex with other components, γ-secretase in yeast microsomes was immunoprecipitated with a PS1 antibody against the loop region (GIL3) (Figure 5). NCT, PS1, and Aph1 were coimmunoprecipitated with wild-type Aph1 or the L30F/T164A mutant (Figure 5, lanes 3 and 4). The recoveries of Aph1 and NCT were similar between the WT and the mutant, suggesting that the mutant Aph1 did not affect γ-secretase complex formation.

### 2.4. Aph1 Mutations Activate γ-Secretase in Mouse Fibroblasts

We analyzed the Aph1 mutations in embryonic fibroblasts obtained from Aph1-a/Aph1-b/Aph1-c triple knockout mice (Aph1 TKO cells). We introduced C99 and Aph1aL genes into Aph1-KO MEF cells using retroviral vectors. The amount of Aph1 L30F/T164A was lower than the amount of wild-type Aph1, but similar amounts of PS1 NTF were detected (Figure 6a, lanes 2 and 3), indicating that Aph1 mutants formed mature γ-secretase complexes. Next, we analyzed the effect of the L30F/T164A mutation on Aβ production. Western blotting showed that the amount of Aβ produced by the L30F/T164A mutant was approximately twice the amount produced by the WT (Figure 6b). We used ELISA to quantify Aβ40 and Aβ42 production by the L30F/T164A mutant; we found that the L30F/T164A mutation resulted in approximately twice as much Aβ40 and Aβ42, compared with the amounts produced by the WT (Figure 6c). The ratio of long Aβ42 with Aβ40 was similar between the WT and mutant (Figure 6d).

Next, we analyzed complex formation by co-immunoprecipitation. The γ-secretase complex was isolated from the total membrane fraction using a PS1 antibody; the amount of Aph1 was examined. The amounts of PS1 and Aph1aL in the complex were similar between the WT and L30F/T164A mutant (Figure 6e, lanes 1 and 2), suggesting that the L30F/T164A mutation does not affect complex formation.

## 3. Discussion

In this study, we identified Aph1 mutations that increase γ-secretase activity using random mutagenesis by error-prone PCR and a yeast reconstitution system. Analysis using a yeast reporter system, an in vitro assay using microsomes, and MEF cells showed that the Aph1aL L30F/T164A double mutation increased the ε-cleavage activity of γ-secretase and Aβ production. Previous studies analyzing mutations in Aph1 showed that conserved amino acids in the transmembrane region—Gln83, Glu84, Arg87 (TMD3), Gly122, Gly126 (GxxxG motif, TMD4), His171 (TMD5), and His197 (TMD6)—are important for γ-secretase complex formation and activity [33]. However, these analyses only introduced mutations in targeted regions and amino acids; they did not assess random mutations along the entire length of the protein. To our knowledge, this is the first report of Aph1a mutations that increase γ-secretase activity.

The role of Aph1 as a scaffold in complex formation is well-known [9,34,35]. Because the mutation found in this study may have also affected complex formation, we examined the amount of complex by immunoprecipitation. The amounts of Aph1aL and PS1 fragments (NTF or CTF) in the complex were identical in the Aph1aL WT and mutant in yeast and MEF cells. Since the endoproteolytic fragments of PS1 reflect the formation of the full complement active γ-secretase complex with NCT and Pen2, these results suggest that the mutation directly affected activity, rather than complex-forming ability.

Cryo-EM analysis of the γ-secretase structure revealed that the Leu30 of Aph1 is located on the amino-terminal side of TMD1 of PS1, while Thr164 is close to the carboxy-terminus of PS1 (Figure 7). The conformation of TMD1 of PS1 changes upon binding E2012, a phenylimidazole-type GSM; it is predicted to undergo piston movement [36]. Therefore, we predict that the L30F mutation in Aph1aL affected the conformation of TMD1 in PS1, resulting in altered activity. The carboxy terminus of PS1 forms substrate-binding sites in cooperation with the hydrophilic loop 1 (HL1) [24]. The distance between the oxygen atom of Thr164 of Aph1 and the oxygen atom of Tyr466 of PS1 is predicted to be 3.5 Å, suggesting that a hydrogen bond (H...O distance, approximately 1.8 Å) is formed in these side chains (Figure 7b). The T164A mutation may eliminate the hydrogen bond with PS1 Y466, resulting in flexible movement of the carboxy terminus of PS1. We propose that Aph1 mutations regulate protease activity through conformational changes in PS1.

This study revealed that Aph1aL may be actively involved in γ-secretase catalytic activity. We analyzed the production of different Aβ species and found that the ratio of long Aβ species (Aβ42 and Aβ43) with Aβ40 was similar between the Aph1aL WT and L30F/T164A double mutant. This finding will lead to a detailed understanding of the generation of Aβ molecular species (Aβ42 and Aβ43) in AD, as well as new γ-secretase inhibitors or modulators targeting Aph1. In a future study, it will be interesting to measure ROS or other biomarkers in apoptotic pathways to analyze the neurotoxicity caused by the Aph1aL mutant.

As for the γ-secretase mutations, more than 150 mutations in PS1 are associated with familial Alzheimer’s disease (FAD) [1]. Recent genome-wide association studies identified Aph1b as an Alzheimer risk gene, and Aph1b T27I is likely the single causal variant [37]. In addition, SNPs in a promoter region of the Aph1a gene was associated with risk for developing sporadic Alzheimer’s disease in a Chinese population [38]. In acne inversa, also known as hidradenitis suppurativa, loss of function mutations were identified in PS1, NCT, and Pen2, but not in Aph1 [39]. Thus far, the Aph1a L30F and T164A mutations in this study have not been found in human SNPs (gnomAD, https://gnomad.broadinstitute.org/gene/ENSG00000117362?dataset=gnomad_r2_1; jMorp by Tohoku Medical Megabank Organization for Japanese database, https://jmorp.megabank.tohoku.ac.jp/202112/variants/by-gene/APH1A; both accessed on 29 December 2021) or somatic mutations in cancer (https://cancer.sanger.ac.uk/cosmic; accessed on 29 December 2021). We speculate that SNPs in the Aph1a coding region may affect the accumulation of Aβ and may have causative or protective effects in Alzheimer’s disease.

## 4. Materials and Methods

### 4.1. γ-Secretase Reconstitution and Reporter Assays

To reconstitute γ-secretase in yeast, human PS1, NCT, Aph1-1aL, FLAG-Pen2, APP-based substrate C55 (amino acids 672–726 from human APP700), and Gal4p were cloned into their respective vectors, as described previously [40]. PS1 and NCT were cloned in pBEVY-T [41]; Pen2 and Aph1-1aL were cloned in pBEVY-L [41]. C55 or C55-Gal4p (C55 ligated with the *SUC2* signal peptide sequence and the *GAL4* gene) were cloned in p426ADH [42]. These recombinant plasmids were transformed into *Saccharomyces cerevisiae* strain PJ69-4A [43]. Random mutations in Aph1aL were introduced by error-prone PCR as described previously [40]. The expression levels of Gal reporters *HIS3* and *ADE2*, as well as *lacZ*, were estimated via cell growth on SD-LWHUAde plates and β-galactosidase activity. β-Galactosidase assays were performed as described previously [40]. Harvested yeast cells (1 × 10^7^) were lysed using glass beads in 30 μL lysis buffer (20 mM Tris-Cl, pH 8.0, 10 mM MgCl_2_, 50 mM KCl, 1 mM EDTA, 5% glycerol, and 1 mM DTT) with a protease inhibitor mix. Cell lysates were centrifuged for 10 min at 15,000× *g*, and the supernatants were used to measure β-galactosidase activity and protein concentrations using the Bradford protein assay (Bio-Rad, Hercules, CA, USA).

### 4.2. In Vitro γ-Secretase Assays Using Yeast Microsomes

Yeast microsomes were prepared as described previously [30]. The microsomal fraction was suspended in γ-buffer (50 mM PIPES, pH 7.0, 250 mM sucrose, and 1 mM EGTA) and subjected to γ-secretase assays as described previously [30]. Microsomes containing 40 μg of protein were solubilized in the presence of 1% CHAPSO on ice for 1 h, and the mixtures were diluted 4-fold in γ-buffer containing protease inhibitors: 50 μM di-isopropyl fluorophosphate, 50 μM phenylmethylsulfonylfluoride, 0.1 μg/mL N^α^-*p*-tosyl-l-lysine chloromethyl ketone, 0.1 μg/mL antipain, 0.1 μg/mL leupeptin, 100 μM EGTA, 1 mM thiorphan, and 5 mM phenanthroline. Where indicated, we added 0.1% phosphatidylcholine (PC) to enhance the activity. The mixtures were incubated at 37 °C and the protein fractions were recovered by a chloroform/methanol (2:1) extraction. Production of Aβ and expression of γ-secretase were detected by immunoblotting after gel electrophoresis. To detect total Aβ, samples were subjected to conventional 16.5% Tris/Tricine SDS-PAGE [44]. To identify the Aβ species, the samples were separated on 10% polyacrylamide gels containing 8 M urea [44]. The blots were developed using an enhanced chemiluminescence system, and signal intensities were quantified with an LAS-4000 image analyzer (Fujifilm, Tokyo, Japan).

### 4.3. Aβ Production in Mouse Embryonic Fibroblasts

To reconstruct Aβ production in mouse embryonic fibroblasts, wild-type or mutant Aph1-aL cDNAs were inserted into the *Eco*RI site of the pMXs-puro plasmid [45]. The retroviral plasmids were transfected into the packaging cell line PLAT-E, as described previously [46]. PLAT-E cells were cultured for 24 h in Dulbecco’s modified Eagle’s medium containing 0.7% polyethyleneimine and 500 μg/mL of plasmid DNA. After 48 h, conditioned media were filtered through membranes with a pore size of 0.45 μm and used as viral stocks. Recombinant retroviruses encoding wild-type or mutant Aph1 were transiently transfected into mouse embryonic fibroblasts lacking Aph1-a, Aph1-b, and Aph1-c (Aph1 triple knockout (Aph1TKO)) [47]. For transfection, cells were cultured with viral stocks containing 5 μg/mL polybrene. After 6 h, the viral stocks were replaced with fresh medium lacking viruses. After incubation for 24 h, conditioned media and cells were collected. To measure the amounts of secreted Aβ species, the collected media were subjected to two-site enzyme-linked immunosorbent assays (ELISAs) with anti-Aβ antibodies, BNT77/BA27 for Aβ40, and BNT77/BC05 for Aβ42 and Aβ43, respectively [48], or immunoblotting analysis [44].

### 4.4. Immunoprecipitation of γ-Secretase

Yeast microsomes (80 μg protein) or the total membrane fraction of MEFs collected from a 10 cm dish were solubilized with IP buffer (1% CHAPSO, 50 mM HEPES, pH 7.4, 150 mM NaCl, 2 mM EDTA, and protease inhibitor cocktail (Sigma, MO)) on ice for 1 h, then centrifuged at 100,000× *g* for 30 min at 4 °C. The supernatant was incubated with primary antibody (1:300) overnight at 4 °C, then incubated with protein A-Sepharose beads for 1 h at room temperature (GE Healthcare, Buckinghamshire). Subsequently, the immunoprecipitants were washed with IP buffer and subjected to immunoblotting.

### 4.5. Antibodies

We used the following antibodies for immunoblotting and immunoprecipitation: monoclonal antibodies against the Aβ N-terminal (82E1; IBL, Gunma, Japan), tubulin (DM1A; Sigma-Aldrich, St. Louis, MO, USA), FLAG (M2; Sigma-Aldrich), and Aph1aL carboxyl terminus (245–265) (Poly18231; Biolegend); and polyclonal antibodies against NCT (AB5890; Chemicon, Temecula, CA, USA), human PS1 amino-terminus (G1NR), and human PS1 loop (GIL3) [35].

## Figures and Tables

**Figure 1 ijms-23-00507-f001:**
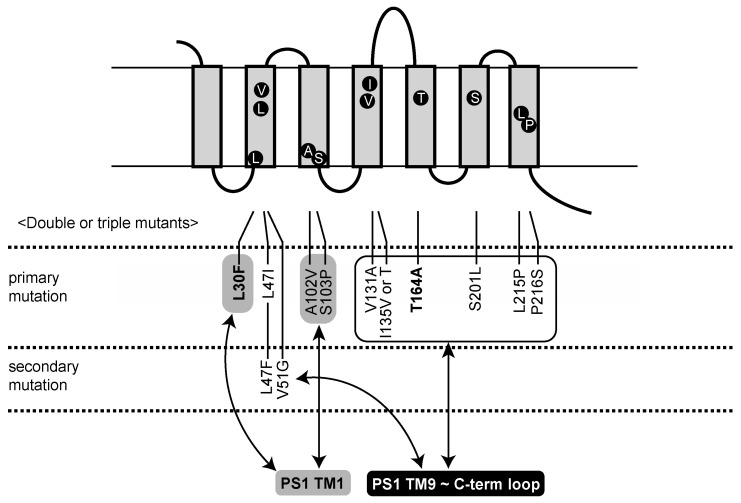
Aph1 mutants active in the absence of NCT map to transmembrane domains. Mutations in double or single mutants are indicated in the model with the seven transmembrane domains (TMDs) of Aph1. Mutations active as single mutants are shown as primary mutations; mutations active only in combination are shown as secondary mutations. The Aph1 mutations are in close proximity to PS1 TMD1 or TMD9 connected to the C-terminal loop, as indicated by arrows. L30F/T164A mutations, which activated γ-secretase, are shown in bold.

**Figure 2 ijms-23-00507-f002:**
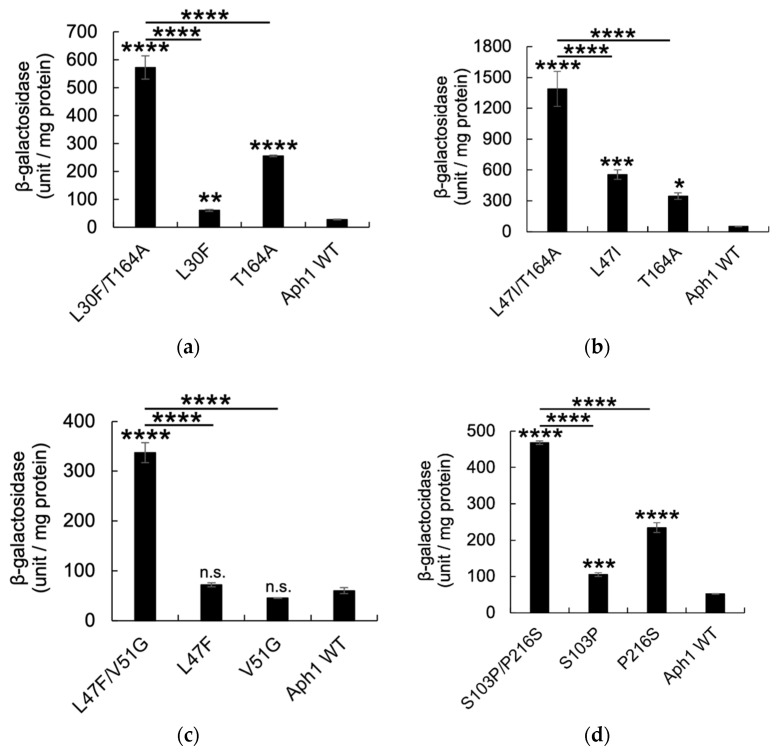
β-Galactosidase activity of Aph1 active mutants in the absence of NCT. Wild-type (WT) or mutant Aph1 was introduced into yeast with PS1, FLAG-Pen2, and APP_C55_-Gal4p; Gal4p cleaved from APP_C55_-Gal4p activates the reporter gene, *lacZ*, encoding β-galactosidase. β-Galactosidase activity was assayed in cells containing Aph1 double and single mutants (**a**–**g**). One unit of β-galactosidase activity corresponds to 1 nmol *O*-nitrophenol β-galactoside hydrolyzed per minute; it is expressed as units/mg lysate protein. Activity was normalized as relative to the activity of control cells without NCT (52.0 units/mg protein). Results from three independent assays are shown with their standard deviations. Statistical analyses were performed using one-way analysis of variance (ANOVA) followed by Dunnett’s multiple comparison test. Asterisks on each bar indicate *p* < 0.001 (****), *p* < 0.001 (***), *p* < 0.01 (**) or *p* < 0.05 (*) (**a**–**g**).

**Figure 3 ijms-23-00507-f003:**
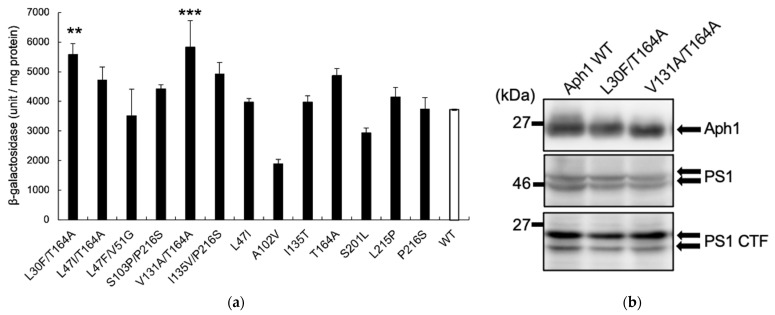
β-Galactosidase activity of Aph1 active mutants in the γ-secretase complex. (**a**) β-Galactosidase activity was assayed in cells expressing the Aph1 wild-type (WT) or active mutants, PS1, FLAG-Pen2, and nicastrin. The experimental conditions were identical to Figure 2. (**b**) Expression of WT or Aph1 mutants and PS1 in lysates were analyzed by immunoblotting using specific antibodies. Endoproteolysis of PS1 was detected by the production of mature fragment PS1 CTF. Asterisks on each bar indicate *p* < 0.01 (**) or *p* < 0.001 (***) (**a**).

**Figure 4 ijms-23-00507-f004:**
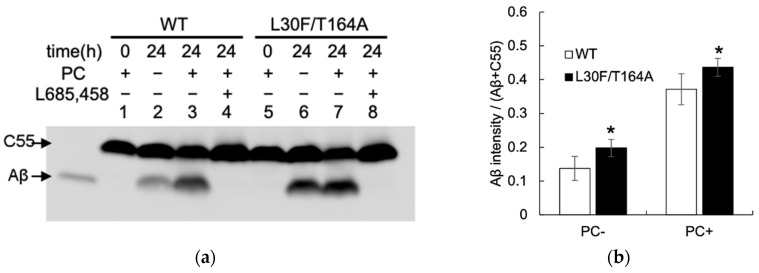
Aph1 mutations activate γ-secretase in yeast microsomes. Microsomes were prepared from yeast transformants expressing Aph1 WT or Aph1 L30F/T164A mutants, PS1, NCT, FLAG-Pen2, and the C55 fragment of APP; they were then subjected to γ-secretase assays. CHAPSO-solubilized microsomes (40 μg protein) were incubated at 37 °C for 24 h in the presence or absence of phosphatidylcholine (PC) (0.1%) and γ-secretase-specific inhibitor (L685, 458). (**a**,**b**) Total Aβ production was analyzed by immunoblotting using the antibody 82E1. (**c**,**d**) Aβ species were separated by urea/SDS-PAGE and analyzed by immunoblotting with 82E1. Synthetic Aβ38, Aβ40, Aβ42, and Aβ43 (30 pg) were loaded as controls in the left-most lanes (**a**,**c**). Amount of Aβ (**b**,**d**) and the ratio of Aβ species (**e**) were quantified from three independent assays; data are shown with standard deviations. Statistical analyses were performed using one-way ANOVA followed by Dunnett’s multiple comparison test. Asterisks on each bar indicate *p* < 0.01 (**) or *p* < 0.05 (*) with respect to Aph1 WT (**b**,**d**,**e**).

**Figure 5 ijms-23-00507-f005:**
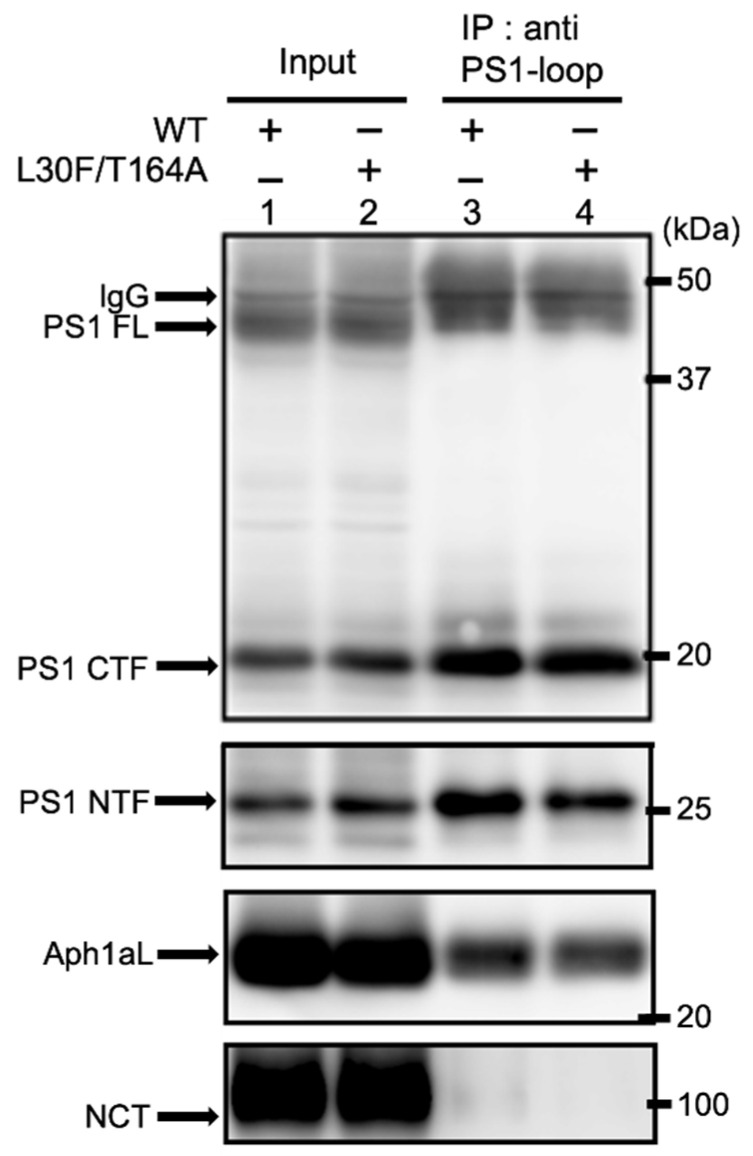
Aph1aL L30F/T164A mutant forms γ-secretase complexes in yeast. Microsomes were prepared from yeast transformants as in Figure 4 and solubilized with buffer containing 1% CHAPSO. The γ-secretase complex was purified from extracts using an antibody against the PS1-loop region (G1L3) or rabbit pre-immune serum, as indicated. Immunoprecipitants and the input fraction were analyzed by immunoblotting using specific antibodies. The input represents 25% of the microsomal extract.

**Figure 6 ijms-23-00507-f006:**
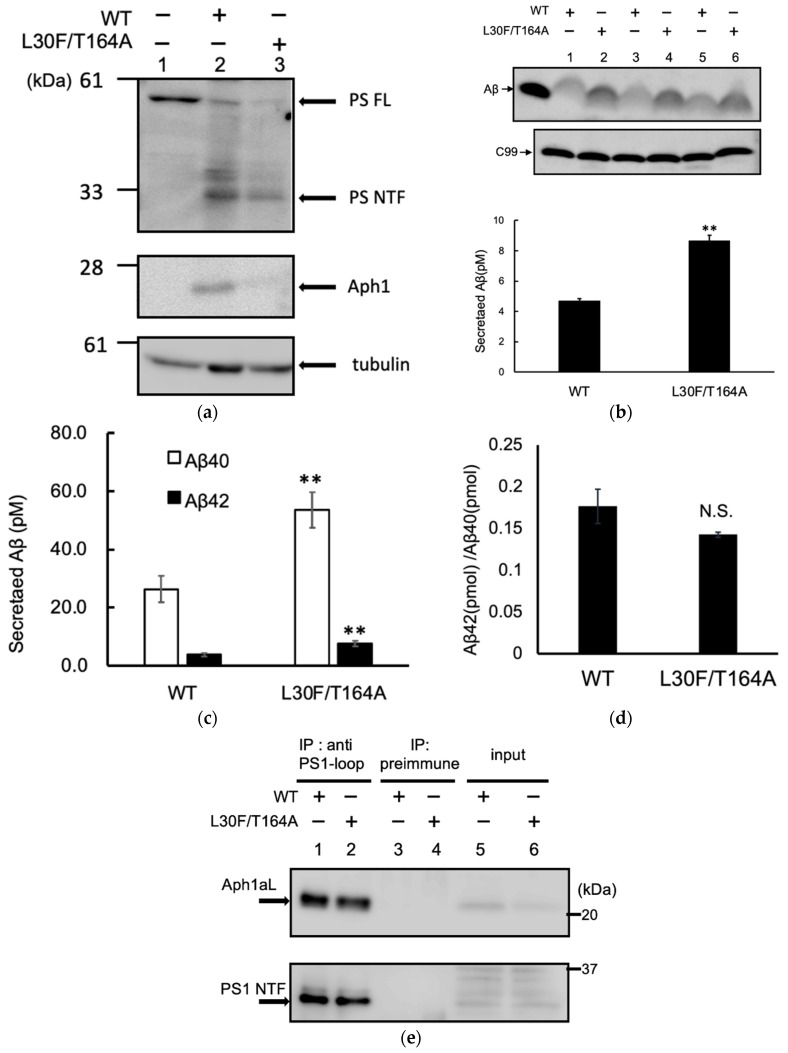
Production of Aβ by Aph1 mutants in mouse embryonic fibroblasts. Using the retroviral system, Aph1 knockout cells were transfected with Aph1 WT or Aph1 L30F/T164A. Cells were harvested; extracts were then analyzed by immunoblotting with the indicated antibodies (**a**). Endoproteolysis of PS1 was detected by the production of mature fragment PS1 NTF. Secretion of Aβ peptides (**b**,**c**). After transfection, the media was recovered, and the total secreted Aβ was analyzed by immunoblotting (**b**). The amount of secreted Aβ was quantified from three independent assays and is shown with standard deviations (**b**). The amounts of Aβ species (**c**) and the ratio of Aβ42 with Aβ40 (**d**) were quantified by ELISA (*n* = 11; data represent the mean ± SD). Asterisks on each bar indicate *p* < 0.01 (**) with respect to the Aph1 WT by the Student’s *t* test (**b**–**d**). The γ-secretase complex was purified from cellular extracts using an antibody against the PS1-loop region (G1L3) or rabbit pre-immune serum, as indicated (**e**).

**Figure 7 ijms-23-00507-f007:**
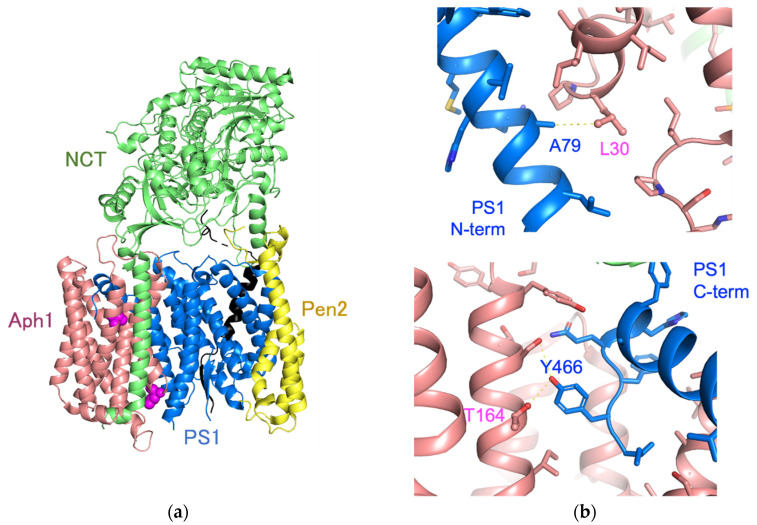
Aph1 mutations in the structure-based model. A structure-based model of the transmembrane core of human γ-secretase (Protein Data Bank code 5A63) is shown with a sphere containing the following residues: Leu30 and Thr164 (purple), PS1 (blue), Aph1aL (pink), NCT (yellow-green), Pen2 (yellow), and APP (black) (**a**). Leu30 and Thr164 of Aph1 are in close proximity to Ala79 and Tyr466 of PS1, respectively (**b**).

**Table 1 ijms-23-00507-t001:** Screening for Aph1 mutants active without NCT. The growth of cells with APP-Gal4, PS1 (WT or S438P), and Pen2 were analyzed on SD-LWHUAde medium after 3 days at 30 °C. “+++” represent full growth, cells formed colonies (>1 mm); “++” represent partial growth with colonies (>0.5 mm); and “-” represent no growth. 1.1 × 10^6^ Cells were screened.

Aph1 Mutants	Growth with PS1S438P	Growth with PS1WT	No. of Clones
L30F/T164A	+++	-	1
L47I	+++	-	4
L47I/T164A	+++	-	1
L47F/V51G	+++	-	1
A102V	+++	-	1
S103P/P216S	+++	-	1
V131A/T164A	+++	-	1
I135V/P216S	+++	-	1
I135T	++	-	1
T164A	+++	-	3
S201L	++	-	1
L215P	+++	-	2
P216S	+++	-	1

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
