# Peer review of "Specific Mutations in Aph1 Cause γ-Secretase Activation"

_ijms, 2022, doi:10.3390/ijms23010507_

Round 1

Reviewer 1 Report

Authors presented the possible involvements of Aph1aL in regulating γ-secretase catalytic activity, especially in the production of Aβ42 and Aβ43. 

Since Aβ42 and Aβ43 are minor species and other mechanisms are involved in the following process, ratios with Aβ40 need to be presented. 

Next, interactions of Aph1 with PEN2 needs to be discusses. 

Lastly, it will be interesting to measured ROS and other biomarkers along the apoptotic pathways.

Reviewer 2 Report

This is a good followup research work after the group's published work about establishing a γ-secretase assay in yeast. Here the authors identified several mutants of γ-secretase can enhance its protease activity from different aspects including a yeast reporter system and cryo-EM analysis etc. The work is of great interest to the readership in IJMS and MDPI publisher and fill the gaps of our knowledge of neurodegenerative diseases. The manuscript is good to be accepted after the minor revision. 

  1. In supporting information, please add the figure caption to each figure for better comprehension for the readers.
  2. For Figure 5 in the main context, please specify the unit of the ladder. 
  3. In the discussion, the authors can provide more detailed discussion of the mutations investigated in the manuscript. For instance, how often the mutation occurs in the γ-secretase to highlight the significance of this research work. 
